# Measuring Inadequacy in Compensation for the Compulsory Acquisition of Land: Evidence from Bengaluru, India

**Jyoti Shukla and Piyush Tiwari \***

Faculty of Architecture, Building and Planning, University of Melbourne, Melbourne 3010, Australia; jyoti.shukla@unimelb.edu.au

\* Correspondence: piyush.tiwari@unimelb.edu.au

**Abstract:** Taking inspiration from the longstanding problem of inadequate compensation for the compulsory acquisition of private land for a public purpose, this research estimates the economic value of (i) future development potential of land or 'hope value'; and (ii) depreciation in property value due to acquisition notification or 'blight'. Using empirical data on property transactions conducted in the Bengaluru Mysore Infrastructure Corridor (BMIC) project area and registered with sub-registrar offices in Bengaluru India during 2007–14, this research innovatively combines the duration model and hedonic price model to estimate the above. Results indicate that the current mechanisms for compensation are inadequately compensating for the land. The loss of the hope value ranges between 2.39 to 8.35 times the market value of agricultural land in 2006 and loss due to blight is approximately 31 percent. Compensating for these losses ex-ante should induce fairness in the compulsory acquisition process and reduce arbitrariness in the valuation of essential components of a compensation package, thus unburdening the valuation responsibilities of the legal institutions. These findings empirically support the argument of payment of additional monetary compensation to the market value of land and provide a rational measure of the same.

**Keywords:** compulsory land acquisition; hope value; blight; fair compensation; India

## 1. Introduction

Procurement of land for large infrastructure and development projects through market mechanisms involves huge transaction and information costs, making the whole process expensive and time-consuming. In India, the lack of accessible land records further extenuates the situation, as it is difficult for market participants to ascertain land ownership, making the negotiation difficult. The consequence has been that the promoters of such projects have eschewed market negotiations for land procurement in favor of the use of compulsory acquisition powers of the state. These powers of the state, legally backed by the provision of enabling law, allow government or its department to acquire a private interest in land (land rights of individuals) for public purposes.

However, the compulsory acquisition process is strongly resisted by the affected private landowners. An avid reader may refer to Shukla [1] for a detailed understanding of the process of compulsory acquisition in India and issues of unfairness. As Rao [2,3,4] discussed, these landowners bear the loss of many functionings[1] associated with land that the present mechanisms of compensation (or resettlement) do not consider. Aggrieved landowners, who often undergo more losses than the compensation received [5,6], seek opportunities for negotiation, which gives rise to legal disputes between the landowners and acquirers. There is ample empirical evidence from many parts of the world, including India, to suggest that the majority of legal disputes for compensation conclude in favor of the landowner and upward improvement in compensation is commonly observed [7,8]. Landowners lose additional money and time in accessing the judicial system[2] to negotiate

over arbitrary components of financial compensation which usually include severance; injurious affection; disturbances; and blight (explained under Section 2).

This research does not question the use of powers of compulsory acquisition and instead argues for a fair compensation that can take account of 'hope value' and 'blight' currently remain uncompensated in many countries, including India. Negative impact of 'blight' and the problem of measuring the impact is specific to the case of compulsory acquisition of land. However, the problem of estimating the 'hope value' is widely acknowledged in the valuation profession [9]. For example, global valuation standards by IVSC [10] and RICS [11] consider it necessary to take account of the likelihood of land use/zoning to change in the future. Thus, while feeding into the bigger objective of designing a fairer compensation mechanism, this research makes methodological contribution into estimation of 'hope value'.

The paper asks the following specific questions:

(i)     how can the economic value of a probabilistic potential for the development of land caused due to a public infrastructure project be measured?

(ii)    how can the negative impact of land acquisition notice on land values be measured?

This paper finds answers to the above questions using the case of the Bangalore-Mysore Infrastructure Corridor (BMIC) project, which is a typical example of a Public-Private Partnership (PPP) road infrastructure project in India.

The rest of the paper is structured as follows: Section 2 summarizes literature on components of compensation and inadequacies. Section 3 presents a brief overview of the study area of BMIC project and its relevance for this research. Section 4 explains the methodology and discusses the theoretical basis for the hedonic model. Section 5 discusses the data sets used in this research. Section 6 presents the results. Section 7 discusses the results and Section 8 concludes the paper.

## 2. Literature

Laws across countries legislate compensation on principles of 'just compensation'; 'fair compensation', and 'equity or equivalence' [12]. However, an over-simplified interpretation of 'just' and 'fair' compensation equates the meaning of compensation to the payment of the market value of land and other tangible losses [12]. Under Section 26 of the Right to Fair Compensation and Transparency in Land Acquisition, Rehabilitation and Resettlement Act, 2013, the market value of land in India is defined as the higher among the following: (a) the transaction value specified in the Indian Stamp Act, 1899 (2 of 1899); or (b) the average sale price for similar type of land situated in the nearest village or nearest vicinity area, during the three years immediately preceding the year in which such acquisition of land is proposed.

In addition to the market value, there are additional components of compensation—although, in most cases, the onus to claim lies on the affected landowner. For example, additional components of compensation in the UK, Denmark, US and New Zealand, though described differently, broadly include the market value of land and improvements; severance; disturbance; and injurious affection (refer to Olanrele et al., [12], for details on components of compensation across different countries). In addition to the above losses, the Indian legislature (Land Acquisition Act, 1894) adds to the market value of land in current use another 30% solatium as compensation in lieu of compulsory nature of sale (refer Section 23(2), Land Acquisition Act of 1894). A similar justification is used by other countries like Australia and Scotland to offer additional solatium on top of market value of property particularly when the primary residence of a person is to be acquired.

In India, legal compensation is revised significantly upwards in the latest act of Right to Fair Compensation and Transparency in Land Acquisition, Rehabilitation and Resettlement Act, 2013 (hereafter referred as LARR, 2013), and the cumulative amount is between two to four times the market value of the land plus the value of assets and improvements on land, if any (refer to Schedule 2, LARR, 2013). However, the new act does not explicitly

state the components of compensation and rather cumulates all losses (i.e., market value; severance; injurious affection; and disturbance) into a lump sum compensation of two to four times the market value. Though a good range of financial losses is covered under these heads, many losses (financial and non-financial) remain uncompensated, as discussed below.

Through a survey of affected landowners in India and Scotland, Rao [3,4] identifies financial (and non-financial) losses that are not considered in the current compensation assessment:

1. Hope Value of land: An essential benefit of owning land is financial benefits linked with the future development potential of land [3]. The usual observation of the land market indicates consistent growth in the demand for land and its value over time. Therefore, landowners are optimistic about the advancement of their land's development potential and consequential improvement in its value. This is at times called the 'hope value'. For an avid reader, Grzesik and Źróbek [13] present an analysis of varying interpretations of the 'hope value'. Rao [3] refers to the definition of 'hope value' used by the Lands Tribunal for Scotland to mean the economic value of the hope of some land development, giving it a value beyond its current use, under a no-scheme scenario[3]. This research relaxes this definition of hope value to include the economic value of future development potential of land arising with or without the proposed scheme for which land is to be compulsorily acquired. Section 4.1 explains 'hope value' in detail.

2. Blight due to acquisition notification: Sometimes, there is a time gap between the actual acquisition of land and its notification for acquisition. Landowners may find it challenging to sell or mortgage land when their land is earmarked for acquisition. Blight is defined as the depreciation in property value consequential to a notification for compulsory acquisition for a public project. For example, when a property is earmarked for a futuristic public purpose, its value may reduce due to the likeliness of it being compulsorily acquired in the foreseeable future. This may make it challenging for the owners to sell or mortgage the property at market value. Some countries, like Scotland, compensate for blight. The owners have the opportunity serve a blight notice on the acquiring authority and force the authority to buy their interest at its value before it was affected by blight [14]. Nevertheless, this is not the case in India, and there is no compensation for 'blight'.

Although Scotland and some other countries acknowledge the loss of 'hope value' and 'blight', the onus lies on the landowner to prove their claim [15]. Arbitration process is expensive and time consuming, thus discouraging landowners from undergoing a legal negotiation [15].

As mentioned by Drapikovskyi [9], option pricing models such as the binomial model, the Black-Scholes model and the Samuelson-McKean model are most popularly employed in determining the hope value of land. However, option pricing models are not dynamic and do not take account of the varying probability of change of land uses over time. To overcome this problem, this research innovatively uses the duration model together with the traditional hedonic price model, as discussed under Section 4. Findings from this research measure these losses in the Indian context and provide guidance on methodology using an innovative amalgamation of duration model and hedonic price model in determining the above (apparent) losses that deserve compensation.

## 3. Study Area: Bangalore-Mysore Infrastructure Corridor (BMIC) Project

The Bangalore (Bengaluru)-Mysore Infrastructure Corridor (BMIC) was conceptualized in 1988 as an instrument for infrastructure-led economic growth in Karnataka state (India). The project had two components (i) to develop an expressway connecting the cities of Bangalore and Mysore and (ii) to develop growth centers (townships) along the expressway to direct growth and act as counter magnates to burgeoning population

growth in Bengaluru and Mysore [16,17]. This project is only partially complete after two decades because of numerous controversies around the land acquisition.

The project used a land-based financialization model of funding and therefore acquired land over what was required for the road [18]. A total of 23,846 acres of land is proposed to be acquired for the project from across eight different Talukas (or administrative sub-divisions within a city) of Bangalore North (885 acres), Bangalore South (5089 acres), Ramnagaram (8170 acres), Channapatna (3572 acres), Maddur (481 acres), Mandya (667 acres), Srirangapatnam (4839 acres) and Mysore (173 acres) [19]. Through a series of acquisition notifications between 1996 and 2004, Karnataka Industrial Areas Development Board (KIADB) had notified all land for compulsory acquisition under the Karnataka Industrial Areas Development (KIAD) Act of 1966. The land was successfully acquired for the first phase of the project. This allowed for constructing a peripheral road around Bengaluru city, and a short stretch (12 km) of the expressway opened for the public in June 2006. At this stage of the project, two townships were to be developed. Still, landowners have been strongly resisting the acquisition of their land for townships because they believe that townships are for private gain and not public purpose (for more details on private gains from the project, refer to report by Raj & Angadi [20].

The status of landowners is that their land is notified for acquisition, but the possession is not taken by the KIADB. As per a report by the Comptroller and Auditor General of India [21], in 2016–17, approximately 28,720 acres of land is held by the KIADB under the preliminary notification, of which 63 per cent is pending acquisition. After the preliminary notification for land acquisition, original landowners continue to operate under legal restrictions imposed by the KIAD act on improving, selling or mortgaging land. The Act does not specify any timeline and the time gap between preliminary notification and actual acquisition is uncertain.

Once the land is earmarked for acquisition (under preliminary notification), its value depreciates. Influential market players see this as the opportunity to procure notified land at reduced prices from desperate landowners. Even though there is a risk of losing out land to the acquirers, there is still an opportunity to get the land de-notified [22]. By its power under Section 21 of the Karnataka General Clause Act, the State Government has the authority to cancel final notification orders (issued under Section 28(4) of the KIAD act) at any time before the possession of land by the acquiring agency. The possibility of de-notification encourages strategic purchase of notified land as observed in the BMIC project.

Amidst solid resistance to the project from the public and political parties, the region witnesses strategic buying and selling of land. Over the years, many real-estate developments, particularly residential apartments, have been developed in Bengaluru South, along the expressway, and the value of land in the region has increased significantly [8].

Time delay in the project has revealed the 'hope value' and 'blight' of the land to the original owners, who are now demanding better value for their land as compensation. However, the acquirers are reluctant to pay anything more than the market value of land in its current agricultural use. Delay in land acquisition has also revealed the negative impact of acquisition notification on the property values.

Due to the factors mentioned above, the BMIC project allows us to observe the changes in the development potential of land over time around the project (hope value) and the negative impact of acquisition notice on land values (blight).

## 4. A Theoretical Framework for Estimation: The 'Hedonic' Approach

### 4.1. Defining 'Hope Value' of Land

Figure 1 explains the relationship between the potential for development of land over time (top section), the certainty of development in the present time (middle section), and the impact of the two on the market value of land (lower section). The economic value of the potential of development of land can be broadly categorized under four groups (i) value derived from the current use of land; (ii) value associated with the highest and best

use of land that is legally permissible, physically possible and financially viable in the current time; (iii) speculative value linked with the development potential of land in the future that is observable or foreseeable in the present time; (iii) future value appreciation linked with the development potential of land in the future that is unobservable or non-foreseeable in the present time, but should improve the value of land in the future.

Financial security arising out of using land for its highest and best permissible use is captured well in its market value. The economic value of this function of land can be interpreted in terms of the land's physical characteristics, which add to its productivity and value, such as its area, nearness to the city center, whether it is irrigated or not (if under agricultural use) and so on. Improvement in the development potential of the land is continuous over time but is captured in the market value to the extent the gain is certain in the present time (Figure 1).

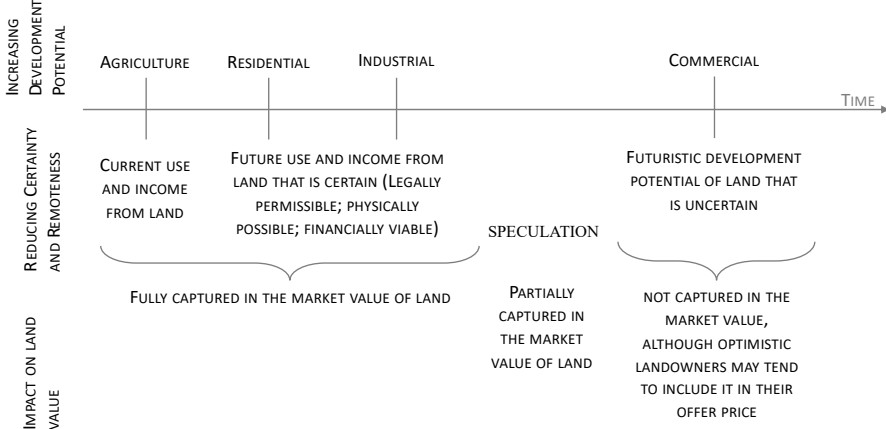

**Figure 1.** Schematic representation of the relationship between the development potential of land; probability of development; and the value of land over time.

While the potential for development improves over time, its impact on the market value depends upon the level of certainty of the development in the present time. Therefore, permissible uses of land are fully reflected in its value. Still, other developments on land for which planning permission is required are uncertain, depending upon the probability of obtaining planning permission, only partially inform the market value. The greater the observability of development in the present time, the more significant is its impact on the market value of the land.

Regarding futuristic and uncertain developments, the 'observability' of the development potential of land may vary among individuals and is, therefore, a matter of subjective judgement. However, due to the speculative nature of the land market, it partially captures the futuristic development potential in the value of land, depending upon the level of certainty associated with the development. It is challenging to distinguish strictly between observable and unobservable developments. The economic value of the financial security arising out of the future development potential of the land is interpreted as below:

The economic value of future development potential of land = probability of development × value of developed land.

This paper refers to the economic value of future development potential of land as 'hope value'. Put another way, hope value is the appreciation in the value of land in the present time due to the futuristic development potential of land that is uncertain. Mathematical interpretation of hope value is expressed in Equation (1) below:

$$\text{Hope Value} = \sum_{t=0}^{\infty} \frac{1}{(1+r)^t}\left\{\left(P_{AA,t} - 1\right)V_{A,t} + P_{AB,t}V_{B,t} + P_{AC,t}V_{C,t} + \cdots + P_{AN,t}V_{N,t}\right\} \tag{1}$$

where $r$ is the inflation rate, $t$ is time, $P$ is the probability of change of land-use from A to N, $V_A$ is the value of land under use A.

### 4.2. Probability of Development or Land-Use Change

The probability of land use changes is estimated using cumulative incidence curves (CIC). CIC is derived from the hazard function for each risk when competing risks are present [23]. For a land 'li', there are four possibilities (let us call them competing risks): land use remains in agriculture use (la), land-use changes to land for development (ll), land-use changes to residential plot (lr) and land-use changes to apartment plot (lapt). There are three CIC possible. The land use, li, could change to ll or lr or lapt. The default is that li stays as la.

For each li, the $CIC_{li}(t_f)$ is given by Equation (2):

$$CIC_{li}(t_f) = \sum_{f'=1}^{f} \bar{I}_{li} = \sum_{f'=1}^{f} \bar{S}(t_{f'-1}) \bar{h}_{li}(t_{f'}) \tag{2}$$

where CIC are cumulative incidence probability.

li is the event type defined by the change in land use to either ll or lr or lapt.

$t_f$ is the time when the land-use changes.

$\bar{I}_{li}$ is the estimated incidence of land-use change from event type li at the time $t_f$.

$S(t_{f-1})$ is the overall probability of no change in land use up to the previous time $(t_{f-1})$.

$\bar{h}_{li}(t_f) = \frac{m_{lif}}{n_f}$ is the land use change at ordered times $t_f$ for event type (li) of interest. $m_{lif}$ are the number of land use changes for type li (either ll, lr or lapt) at time $t_f$ and $n_f$ are the total number of land uses at time $t_f$.

The cumulative probability of land use as agriculture at $t_f = 1 - CIC_{ll}(t_f) - CIC_{lr}(t_f) - CIC_{lapt}(t_f)$.

### 4.3. Alternative Functional Forms for Hedonic Price Function

As per the hedonic hypothesis, goods are valued for their utility-bearing attributes or characteristics. In his seminal paper, Rosen [24] describes differentiated products as a vector of objectively measurable characteristics and explains that "observed product prices and the specific amounts of characteristics associated with each good define a set of implicit or 'hedonic' prices" (p. 34). A simplified interpretation of Rosen's [24] model could be Equation (3):

$$\text{Observed product price or market value} = \sum_{i=1}^{n} \text{amount of characteristic}_i \text{ x implicit price of characteristic}_i \tag{3}$$

The hedonic slopes of land price function are the implicit prices of land attributes. Since we do not have any prior notions about the shape of the hedonic function, we estimate alternative forms of Box-Cox transformations. The most general Box-Cox functional form in the literature is [25,26] Equation (4):

$$P(Z)^{(\tau)} = \beta_0 + \sum_{i=1}^{k} \beta_i Z_i^{\lambda_i} + 0.5 \sum_i \sum_j \gamma_{ij} Z_i^{\lambda_i} Z_j^{\lambda_j} \tag{4}$$

where

$$P(Z)^\tau = [(P(Z))^\tau - 1]/\tau$$

and

$$Z^{(\lambda_i)} = (Z^{(\lambda_i)} - 1)/\lambda_i$$

We estimate the following functional forms

1. $\tau = 1, \lambda = 1, \gamma_{ij} = 0$; Linear
2. $\tau = 0, \lambda = 1, \gamma_{ij} = 0$; Semi − log
3. $\tau = 0, \lambda = 0, \gamma_{ij} = 0$; Log − log
4. $\gamma_{ij} = 0$; Box − Cox~Linear
5. $(\tau, \lambda, \gamma_{ij})$ unrestricted; Box − Cox ~ quadratic

where the βs are the market –determined parameters, λ is a parameter used to transform land characteristics in Box-Cox analysis and τ is transformation parameter for land value (P). Nonlinear methods are used to find optimal values of transformation parameters. The first derivative of land value with respect to land characteristics in the above equation are the implicit prices. In this paper we estimate the Box-Cox model, which allows transformation coefficients τ, $\lambda_i$ and $\gamma_{ij}$ to vary freely.

The linear and log-linear model is estimated using ordinary least squares regression method. Box-Cox models are estimated using maximum likelihood estimation technique.

While the implicit (marginal) prices for various characteristics in a log-linear model can be computed easily by multiplying the estimated coefficient with mean value of dependent variable, its computation for Box-Cox is not straightforward. Since the Box-Cox model has transformed dependent and independent variables the implicit prices need to be computed as follows: first, we need to estimate conditional expectation of P given Z under the assumption that the error term and explanatory variables are independent and by using the smearing technique suggested by Duan [27]. Second, the mean of estimator of P so obtained is used to estimate implicit (marginal) prices. The implicit price takes the following form [25], Equation (5):

$$\text{Implicit (marginal) price} = \beta_i P^{1-\tau} Z_i^{\lambda-1}. \tag{5}$$

The value of this function is calculated at each observation level in the sample. Mean implicit price is then calculated by averaging these values over all observations. Huang and Kelingos [28] explain that the conditional mean function for P depends on the value of τ. If $-1 \le \tau < 0$, the re-transformed P may not have finite expectations and mean of P may not exist [28,29]. If this is the case, implicit prices cannot be calculated. For all other values of τ mean of expected P exists and implicit prices can be calculated.

## 5. Data and Variables

The data for this research is obtained from the land and property sales registration data obtained from the Inspector General of Stamps and Registration, Government of Karnataka. As of February 2016, 34,799 sale transactions were digitally[4] recorded in the Bengaluru region between the years 2006 and 2015. Given the low number of digitally recorded transactions in 2006 and 2015, the period of analysis for this paper is limited to 2007–14. There are a total 33,424 of usable transactions in the database. Each record has information on the transaction date, sale price, land area, land use and location. Four major land-use types could be derived from the data: agriculture (3497 records); vacant land (available for development, 25,206 records); residential use (low rise houses at the urban periphery, 3981 records); residential use (apartments, 740 records). Land-use is observed from the data rather than zoning regulations, as these villages are unplanned and not covered by urban planning regulations. These are essentially the land 'in use' rather than 'permitted' use. However, formal permission for change of use (agricultural to non-agricultural) is required for the apartment development. Given the organic nature of development in villages (except for the apartment development), it is assumed that the improvements on the land (except for apartments) do not add to the value of land. Thus, the transaction value is essentially the value of the land itself. In the case of apartments, land is under shared ownership of apartment owners. Each owner of an apartment has a share in land in proportion to the size of their apartment. Besides land and the built structure, there are other characteristics (such as number of parking, amenities, floor level) that are associated with apartments that influence price of an apartment even within same apartment complex. In the data, the apartment value is reported, which is not the land value. For analysis of apartments in this paper, we need to derive the land value for the whole land parcel on which an apartment complex is built. To extract the value of land from the transaction value of apartment, a hedonic price function is estimated under Section 6.1. The data set contains information on 33,424 sale transactions that have been concluded in 13 notified (those which have received notification for acquisition) and 10 non-notified

(neighboring but not notified for acquisition) villages between 2007 and 2014. Of the total number of transactions, 29 percent are in the notified villages. In analysis, the data on land transactions is supplemented with data from the Census 2011. The Census presents village level demographic, economic and amenities data. For population (total, and marginal social classes—scheduled caste (SC) and scheduled tribe (ST)) we have used Census 2001 and Census 2011 data to interpolate data between the years 2006 and 2011 and extrapolate from 2012 to 2015. An annual geometric growth rate calculated over 2001–11 is used to interpolate or extrapolate. Locational parameters are also observed at the village level and are assumed to remain the same for all observations in that village but vary across villages. These parameters include total geographical area of the village; area under irrigation; non-agricultural area; primary manufacturing activity in the village; infrastructure facilities such as number of public and private health care centers; education facilities; availability of drinking water tap; and drainage availability. Data on locational parameters available in census for the year 2011 is assumed to be consistent during 2007–14.

Alongside agriculture, many acquired villages are involved in non-agricultural primary economic activities. As per Census 2011, 34% villages undertake textile manufacturing as their primary economic activity in addition to agriculture. Manufacturing of medicine and bricks is the primary economic activity for 27 percent and 16 percent of villages, respectively. Greater productivity of land in these villages means higher land values and greater rate of appreciation of land value when compared to purely agricultural land.

## 6. Results

The land transaction database includes transaction of apartments. Information on value of land used for apartments, referred to as the undivided share of land (USL), is crucial to this research. However, this is unobserved. Therefore, we used a hedonic price model to extract the value of USL. The hedonic function considered the transaction value of apartment as a function of USL and other attributes presented in Table 1.

The undivided share of land is defined as the proportionate share of land of an apartment in the land area of the entire apartment complex project. This is calculated by dividing the built-up area or saleable floor area of a single apartment by total built up area or saleable floor area of the entire project multiplied by total land area of the project. Estimated function is then used to predict value of undivided share of land for all observations pertaining to apartment sales in the data. This allows us to estimate hedonic price function for land for all apartment sales. In the second step hedonic functions for land sales have been estimated.

**Table 1.** Hedonic price model estimation for market value of apartments. Dependent variable: Market value of apartment (Log).

| Variables | Coefficients |
|---|---|
| Undivided share of land (log) | 0.451 *** |
|  | (0.041) |
| Number of bedrooms | 0.192 *** |
|  | (0.018) |
| Number of car parking slots | 0.174 *** |
|  | (0.013) |
| Dummy for location of apartment on floor level higher than 4 (if yes = 1; 0 = otherwise) | 0.044 *** |
|  | (0.009) |
| Dummy for the year of transaction—2011 (if yes = 1; 0 = otherwise) | 0.038 ** |
|  | (0.016) |
| Dummy for the year of transaction—2012 (if yes = 1; 0 = otherwise) | 0.119 *** |
|  | (0.015) |
| Dummy for the year of transaction—2013 (if yes = 1; 0 = otherwise) | 0.152 *** |
|  | (0.016) |
| Dummy for the year of transaction—2014 (if yes = 1; 0 = otherwise) | 0.499 *** |
|  | (0.026) |
| Constant | 11.044 *** |
|  | (0.236) |
| Number of Observations | 739 |
| R-squared | 0.750 |

Note: Standard errors in parentheses; *** $p < 0.01$, ** $p < 0.05$. Standard errors are presented in parenthesis. Source: Authors based on Shukla [8].

*6.1. Hedonic Price Modelling of Apartment Sale Value*

The estimated hedonic price model estimation of apartment market value as a function of apartment characteristics and undivided share of land that is associated with the apartment is presented in Table 1. The explanatory variables included in the function are undivided share in land, characteristics of apartment, amenities and time dummies. It is hypothesized that characteristics such as number of bedrooms and amenities such as number of car parking slots have a positive impact on apartment value. All variables have expected signs. The elasticity of undivided share of land is 0.45 implying that 1 per cent increase in undivided share of land for an apartment increases the value by 0.45 per cent. Undivided share of land is correlated with area of apartment but does not correlate with variables included in the function.

The values of undivided share of land for land under apartment use are predicted using hedonic price estimation in Table 1. Instead of the transaction value of apartment in the data, the predicted value of apartment land is used in estimation of hedonic price modelling of land sales in Section 6.2.

*6.2. Hedonic Price Modelling of Land Values*

Land values in a village are a function of land use, social and economic characteristics of village, proximity to town and its influence on land use. Notification for compulsory acquisition negatively impacts that value. Time also plays an important role as it captures the impact of inflation. A hedonic function for land value should include these explanatories in explaining the land values.

Since there is no a priori reason for imposing a functional form for hedonic function for land sales, five alternative functions, discussed earlier under Section 4.3, have been estimated: (i) Linear (ii) Log-Linear (iii) Log-Log (iv) Linear Box-Cox and (v) Generalized Box-Cox. Table 2 presents the results for these models for pooled observations (for all time

periods). The significance level of estimated coefficients is also indicated in Table 2. All variables have a priori expected signs in all functional forms. The Breusch-Pagan test for linear, log-linear and log-log model indicates that the variance is inconsistent and there is heteroscedasticity. Estimation of hedonic function in Box-Cox form overcomes heteroskedasticity in the data as the transformation of dependent and/or independent variables results in unbiased estimators. Variables that assume zero values (dummy variables) and variables that are in 'share' form have not been transformed. The estimates of slope coefficients across various functional forms are not comparable. Table 2 also reports significance of coefficients and Likelihood Ratio test. Literature [25,30] reports implicit prices across various functional forms for Box-Cox, which can be compared. We note that for a generalized Box-Cox form, the transformation coefficients for the dependent and independent variable (land area) are different from each other in magnitude. The problem, however, is that the value of transformation coefficient of dependent variable is in range $-1 \leq \tau < 0$ for both the estimated Box-Cox functions, which makes computation of implicit prices impossible. It may be noted in Table 2 that the transformation coefficients are not very different than zero, Likelihood Ratio test statistics close to $\tau = 0$ , hence a log-log model would be an appropriate functional form. However, there is a concern of heteroscedasticity in estimated log-log function. The problem of heteroskedasticity does not cause bias in the OLS estimates of the coefficients though it tends to underestimate standard errors. In the subsequent analysis and estimation of blight and hope values, we have used estimates from the log-log functional form. Additionally, Table 3 presents estimates of the hedonic price function for land for each of the time-period using the log-log functional form. This helps in understanding the changes in market preferences for characteristics over time. The explanatory variables retained in Tables 2 and 3 are those which are significant at the 5% level.

**Table 2.** Estimation results for alternative hedonic price functions for land value. Standard errors in parentheses: *** $p < 0.01$, ** $p < 0.05$, * $p < 0.1$.

| Variables | (1) Linear | (2) Log-Linear | (3) Log-Log | (4) Box-Cox (LHS) No-Trans | (5) Box-Cox Linear (Both Side) No-Trans | (6) Trans |
|---|---|---|---|---|---|---|
| Area of land transacted (sq.m) | 851.7 *** | 0.000115 *** | 0.769 *** | 2.4x10⁻⁵ | | 1.000 |
| | (6.376) | (1.90e-06) | (0.00348) | (0) | | (0) |
| Non-agricultural area in the village | 2.773e+06 *** | 3.356 *** | 2.648 *** | 0.823 | | 0.00629 |
| (as a share of total geographical area in the village) | (223,063) | (0.0663) | (0.0447) | (0) | | (0) |
| SC Population in the village | −523,133 *** | −1.037 *** | −1.257 *** | −0.261 | | −0.0513 |
| (share of total population in the village) | (161,564) | (0.0480) | (0.0323) | (0) | | (0) |
| ST Population in the village | 301,711 | −0.0365 | −0.699 *** | −0.0276 | | −0.00458 |
| (share of total population in the village) | (258,049) | (0.0767) | (0.0517) | (0) | | (0) |
| Dummy for primary manufacturing activity in the village (bricks): 1 = yes; 0 = otherwise | 909,416 *** | 0.475 *** | 0.479 *** | 0.101 | 0.175 | |
| | (54,162) | (0.0161) | (0.0108) | (0) | (0) | |
| Dummy for primary manufacturing activity in the village (medicine): 1 = yes; 0 = otherwise | 487,675 *** | 0.147 *** | 0.329 *** | 0.0299 | 0.131 | |
| | (54,228) | (0.0161) | (0.0109) | (0) | (0) | |
| Dummy for village notified for acquisition: 1 = yes; 0 = otherwise | −641,440 *** | −0.246 *** | −0.374 *** | −0.0551 | −0.197 | |
| | (43,792) | (0.0130) | (0.00877) | (0) | (0) | |
| Dummy for nearest town from the village being BBMP: 1 = yes; 0 = otherwise | 183,311 *** | 0.695 *** | 0.609 *** | 0.181 | 0.348 | |
| | (40,494) | (0.0120) | (0.00810) | (0) | (0) | |
| Dummy for residential land-use (apartment): 1 = yes; 0 = otherwise | −730,576 *** | 0.370 *** | 1.357 *** | −0.0150 | 0.673 | |
| | (130,642) | (0.0388) | (0.0265) | (0) | (0) | |

| | | | | | | |
|---|---|---|---|---|---|---|
| Dummy for residential land-use (house): 1 = yes; 0 = otherwise | 219,861 ** | 0.181 *** | 1.071 *** | 0.0472 | 0.387 | |
| | (100,798) | (0.0300) | (0.0205) | (0) | (0) | |
| Dummy for vacant land: 1 = yes; 0 = otherwise | −589,398 *** | −0.593 *** | 0.499 *** | −0.132 | 0.138 | |
| | (93,198) | (0.0277) | (0.0193) | (0) | (0) | |
| Percentage of area under irrigation in the village (share of total geographical area in the village) × Transacted property being in agricultural use (dummy) | −602,416 | 0.595 *** | 1.541 *** | 0.176 | 0.408 | |
| | (441,635) | (0.131) | (0.0882) | (0) | (0) | |
| Dummy for the year of transaction—2008: 1 = yes; 0 = otherwise | 290,345 *** | 0.0326 * | 0.122 *** | 0.00372 | 0.0559 | |
| | (65,085) | (0.0194) | (0.0130) | (0) | (0) | |
| Dummy for the year of transaction—2009: 1 = yes; 0 = otherwise | 419,893 *** | 0.0368 * | 0.204 *** | 0.00683 | 0.0993 | |
| | (66,407) | (0.0197) | (0.0133) | (0) | (0) | |
| Dummy for the year of transaction—2010: 1 = yes; 0 = otherwise | 131,703 ** | −0.0338 * | 0.103 *** | −0.00233 | 0.0734 | |
| | (65,483) | (0.0195) | (0.0131) | (0) | (0) | |
| Dummy for the year of transaction—2011: 1 = yes; 0 = otherwise | 127,344 ** | 0.0473 *** | 0.209 *** | 0.0173 | 0.115 | |
| | (57,957) | (0.0172) | (0.0116) | (0) | (0) | |
| Dummy for the year of transaction—2012: 1 = yes; 0 = otherwise | 327,187 *** | 0.338 *** | 0.476 *** | 0.0906 | 0.238 | |
| | (56,269) | (0.0167) | (0.0113) | (0) | (0) | |
| Dummy for the year of transaction—2013: 1 = yes; 0 = otherwise | 517,505 *** | 0.517 *** | 0.645 *** | 0.135 | 0.318 | |
| | (56,786) | (0.0169) | (0.0114) | (0) | (0) | |
| Dummy for the year of transaction—2014: 1 = yes; 0 = otherwise | 830,178 *** | 0.779 *** | 0.911 *** | 0.196 | 0.456 | |
| | (62,325) | (0.0185) | (0.0125) | (0) | (0) | |
| Constant | 358,696 *** | 12.58 *** | 8.031 *** | 6.859 | 5.295 | |
| | (100,707) | (0.0299) | (0.0297) | (0) | (0) | |
| Observations | 33,424 | 33,424 | 33,424 | 33,424 | 33,424 | 33,424 |
| R-squared | 0.404 | 0.568 | 0.805 | | | |
| lamda | | | | | −0.181 *** | |
| | | | | | (0.00523) | |
| theta | | | | −0.108 *** | −0.0576 *** | |
| | | | | (0.00400) | (0.00336) | |
| sigma | | | | 0.168 | 0.222 | |
| | | | | (0) | (0) | |

| Test H0: Box-Cox (LHS) | Restricted log likelihood |
|---|---|
| theta= −1 | −488,977 |
| theta= 0 | −469,602 |
| theta= 1 | −537,299 |
| Test H0: Box-Cox Linear (Both Side) | |
| theta = lambda= −1 | −482,561 |
| theta = lambda= 0 | −457,441 |
| theta = lambda= 1 | −537,299 |

**Table 3.** Year-wise estimation of land value (log-log functional form) (dependent variable is land value). Standard errors in parentheses: *** $p < 0.01$, ** $p < 0.05$, * $p < 0.1$.

| Variables | (1) 2008 | (2) 2009 | (3) 2010 | (4) 2011 | (5) 2012 | (6) 2013 | (7) 2014 |
|---|---|---|---|---|---|---|---|
| Area of land transacted (sq.m) | 0.839 *** | 0.810 *** | 0.733 *** | 0.727 *** | 0.824 *** | 0.810 *** | 0.779 *** |
| | (0.0122) | (0.0134) | (0.0103) | (0.00845) | (0.00776) | (0.00826) | (0.00820) |
| Non-agricultural area in the village (as a share of total geographical area in the village) | 2.527 *** | 3.328 *** | 4.336 *** | 4.652 *** | 3.333 *** | 1.720 *** | 1.513 *** |
| | (0.178) | (0.140) | (0.139) | (0.133) | (0.118) | (0.101) | (0.0856) |
| SC Population in the village (share of total population in the village) | −3.394 *** | −2.161 *** | −1.666 *** | −1.591 *** | −0.966 *** | −0.614 *** | −0.556 *** |
| | (0.111) | (0.101) | (0.102) | (0.0860) | (0.0825) | (0.0773) | (0.0668) |
| ST Population in the village (share of total population in the village) | −4.219 *** | −0.827 *** | −0.864 *** | 0.803 *** | 0.387 *** | −0.558 *** | −0.160 |
| | (0.222) | (0.186) | (0.168) | (0.141) | (0.115) | (0.117) | (0.0999) |
| Dummy for primary manufacturing activity (bricks): 1 = yes; 0 = otherwise | Omitted | Omitted | 0.798 *** | 0.718 *** | 0.657 *** | 0.668 *** | 0.800 *** |
| | | | (0.0320) | (0.0263) | (0.0215) | (0.0246) | (0.0292) |
| Dummy for primary manufacturing activity (medicine): 1 = yes; 0 = otherwise | Omitted | Omitted | 0.458 *** | 0.413 *** | 0.521 *** | 0.509 *** | 0.460 *** |
| | | | (0.0370) | (0.0309) | (0.0258) | (0.0279) | (0.0298) |
| Dummy for notified for acquisition: 1 = yes; 0 = otherwise | −0.298 *** | −0.263 *** | −0.275 *** | −0.459 *** | −0.475 *** | −0.417 *** | −0.323 *** |
| | (0.0309) | (0.0266) | (0.0306) | (0.0240) | (0.0187) | (0.0196) | (0.0225) |
| Dummy for nearest town being BBMP: 1 = yes; 0 = otherwise | 0.839 *** | 0.999 *** | 0.551 *** | 0.668 *** | 0.396 *** | 0.303 *** | 0.0544 ** |
| | (0.0263) | (0.0253) | (0.0275) | (0.0213) | (0.0188) | (0.0219) | (0.0247) |
| Dummy for residential land-use (apartment): 1 = yes; 0 = otherwise | Omitted | Omitted | 1.012 *** | 1.199 *** | 1.121 *** | 1.122 *** | 1.039 *** |
| | | | (0.0711) | (0.0733) | (0.0507) | (0.0583) | (0.0877) |
| Dummy for residential land-use (house): 1 = yes; 0 = otherwise | 1.619 *** | 1.490 *** | 0.346 *** | 0.570 *** | 0.770 *** | 1.026 *** | 0.801 *** |
| | (0.0783) | (0.0720) | (0.0573) | (0.0584) | (0.0476) | (0.0485) | (0.0458) |
| Dummy for vacant land: 1 = yes; 0 = otherwise | 1.369 *** | 1.123 *** | 0.0473 | 0.270 *** | 0.133 *** | 0.321 *** | 0.247 *** |
| | (0.0726) | (0.0690) | (0.0531) | (0.0547) | (0.0451) | (0.0464) | (0.0430) |
| Percentage of area under irrigation in the village (share of total geographical area in the village) × Transacted property being in agricultural use (dummy) | 4.403 *** | 3.998 *** | 0.254 | 1.211 *** | −0.676 *** | 0.853 *** | 0.560 *** |
| | (0.304) | (0.323) | (0.249) | (0.246) | (0.214) | (0.203) | (0.210) |
| Constant | 7.521 *** | 7.326 *** | 8.618 *** | 8.436 *** | 8.515 *** | 8.678 *** | 9.346 *** |
| | (0.111) | (0.111) | (0.0732) | (0.0708) | (0.0641) | (0.0664) | (0.0622) |
| Observations | 2364 | 2250 | 2822 | 5055 | 6806 | 7004 | 4316 |
| R-squared | 0.864 | 0.823 | 0.851 | 0.799 | 0.814 | 0.777 | 0.807 |

As shown in Table 3, while the coefficients of "notified" dummy have been negative in all years, as expected, their magnitudes have declined over time until 2010 and increased since then. Three possible explanations may be accorded. First, since the transactions were permitted in notified land till these were formally acquired, and the value of compensation was still negotiable, the gains from the road development percolated to 'notified' land as well, thereby increasing values particularly before 2010 and after 2013. The increase in the value of notified land, after 2013, was much higher as the initial base was quite low compared to 'non-notified' land. The second possible reason is that there

was an increase in opportunistic investors/buyers who were willing to pay higher prices in anticipation of being able to negotiate better compensation from government in future when the land would be formally acquired, and compensation settled. Third, the development activity has created other opportunities for the use of land. The higher price paid for notified land in later years also depicts the ability of the buyer to use the land for other profitable purposes than agriculture as with development, opportunities for brick manufacturing and other commercial activities arose. There is also a possibility that the developer bought some of the land at a higher price than the government acquisition price to avoid resistance.

The results indicate that the transaction value at means, for a given year, is 31 per cent lower for land that is notified for acquisition, compared to the land in villages that are not acquired, ceteris paribus. As mentioned earlier, blight due to notification is possible to be observed in this case because land has not yet been formally taken and transaction of notified land was permitted under the KIAD act of 1966. Notice for acquisition causes blight even though the amount of compensation value offered on each land parcel initially is revised a few times before land is formally acquired and compensation settled. As seen from Table 3 year-wise models, the coefficient of notification dummy in earlier periods are higher before stabilizing in later periods, as the project progresses and benefits from the project offset blight to some extent.

As for the size of land transacted, 1 per cent increase in land area increases the transaction value by 0.77 per cent (Table 2, log-log function), and the area elasticity has remained between 0.73 and 0.84 during 2007–14 (Table 3). Given that these are largely agrarian settlements, irrigated land fetches a higher value compared to rain fed or unirrigated land. The sign for the coefficient of share of irrigated area in total agriculture area is positive and significant. Land value is higher in villages undertaking non-agricultural economic activities, that is manufacturing textile, medicine and bricks. This is indicated by the positive coefficient of non-agricultural land area as a percent of total geographical area of the village. Villages that engage in brick manufacturing have higher value of land than others, as indicated by positive and significant coefficient for the dummy variable (bricks). These are the villages which are closer to BBMP and where large apartment building activity has commenced particularly since 2011 when the ring road component of expressway was completed. Landowners have sold their land at higher values as the demand for brick manufacturing to meet the demand of apartment construction activity in the village has increased. Potential for economic activities also has a significant impact on land values. The main economic activity in most villages prior to 2006/2007 was agriculture. Construction activities that began after the completion of road in 2011 led to the sale of agricultural land by some landowners to brick manufacturers in many of the notified villages. The prospect of losing land had extenuated this situation and the value realized for land for brick manufacturing activity was higher than for agricultural activity.

Land in villages that have become part of Greater Bengaluru Municipal Corporation (BBMP) is valued higher. This reflects the premium attached to being part of metropolitan Bengaluru and being served by Greater Bengaluru Municipal Corporation.

Transaction value is further depreciated due to a higher share of landowners from the Scheduled Castes (SC) and Scheduled Tribe (ST) in the village as indicated by the negative signs for variables per cent SC population and per cent ST population. Greater per cent share of SC and/or ST population in village implicitly implies a higher number of land owners from these socially marginalized groups. Longstanding protective legal restrictions on sale of SC/ST owned land to persons who are not SC/ST has probabilistically lowered the value of their land. As discussed earlier, many of the notified villages have higher SC/ST population than non-notified villages.

Land use also impacts sale value. The coefficients of land use dummies and their signs are as expected. Land for apartments is valued higher than other uses as evidenced from coefficients of land use dummied. The magnitude of coefficient for apartment use is followed by house and vacant site with agriculture as the base. This hierarchy has

remained stable over time. Another set of variables that demonstrate loss of hope value is the coefficients for observed land use. Many of the land transactions on non-notified land have been for uses other than agriculture. The coefficient of land use—apartment is highest, followed by land use—house, land use—shop and land use—agriculture. The owners of land in notified villages lose the opportunity of transacting their land for higher value uses.

### 6.3. Probability of Change of Use

Urbanization, economic growth and development of infrastructure projects cause changes to the land use. As discussed earlier, change in use could also occur in the absence of formal planning processes, as is seen in case of villages around BMIC corridor. The valuation of land for fair compensation purposes in a compulsory purchase for public infrastructure development needs to account for the potential for change to other 'higher' uses in the future. Using competing risk survival analysis as discussed in Section 4.2, the cumulative probabilities of conversion to different types of land uses have been calculated. These are presented in Table 4.

**Table 4.** Cumulative probabilities of land use conversion.

|  | Cumulative Probability—Developable Land | Cumulative Probability—Residential Plot | Cumulative Probability—Land for Apartment | Cumulative Prob—Agriculture Use |
|---|---|---|---|---|
| 2007 | 0.0425 | 0.0039 | 0.0001 | 0.9615 |
| 2008 | 0.0825 | 0.0065 | 0.0002 | 0.9242 |
| 2009 | 0.1188 | 0.0122 | 0.0009 | 0.8943 |
| 2010 | 0.1568 | 0.0215 | 0.0121 | 0.8768 |
| 2011 | 0.2377 | 0.0306 | 0.0151 | 0.8080 |
| 2012 | 0.3474 | 0.0385 | 0.0151 | 0.7063 |
| 2013 | 0.4521 | 0.0509 | 0.0151 | 0.6139 |
| 2014 | 0.5172 | 0.0586 | 0.0151 | 0.5566 |

Probabilities calculated in Table 4 are used for estimating the hope value, as discussed in Section 6.4.

### 6.4. Blight and Hope Value

Estimated impacts of notification for compulsory purchase (or blight) and improvement in the development potential (or hope value) on the transaction value for land are presented in Table 5. The results are presented for three types of villages: (i) the village is predominantly agriculture, (ii) the village besides agriculture also has textile manufacturing activity and (iii) the village engages in agriculture and brick manufacturing. Manufacturing activities compete with agriculture with the consequence that overall land values rise. The value of land in each of the future year is calculated as the weighted average of mean land values for agriculture, developable land, land for residential and land for apartment use. The weights are probabilities calculated in Table 4. 'Hope' value, as discussed in Section 4.1 (Equation (1)), is the present value of the weighted sum of the difference of the mean value of agriculture land for a particular year in agriculture dominated village and the value of land under the scenario that there is conversion to other uses happening in the village. A discount rate of 10% is assumed to calculate the present value. The hope value presented in Table 5 is a multiplier of the mean value of agriculture land in agriculture dominated village in 2006.

Results indicate that the value of notified land is 31 per cent lower than non-notified land, in any given year. Reduction in value is the difference in predicted value of "notified" and "non-notified" land at means in the same year, expressed as a percent of "non-notified" land.

**Table 5.** Mean Blight (% of market value of land in non-notified villages) and Hope Value (in multiples of market value of agriculture land in non-notified agriculture dominated village in 2006).

| Village Level Activities | Change in Land Value |
| --- | --- |
| Agriculture (notified for acquisition) | −31% (Blight) |
| Hope value (in multiples of agriculture land value in 2006, at means) | |
| Agriculture + Medicine manufacturing | 6.17 times |
| Agriculture + Brick manufacturing | 8.35 times |
| Agriculture | 2.39 times |

Source: Authors' own calculations.

As discussed earlier in Section 4.1, the development potential of land is observed through the change of use to higher-value use or through improvement in the land market. Taking base value of non-notified land under agricultural use, in an agricultural village, the hope value ranges between 2.39 to 8.35 times the market value, depending upon the use of land and village level activities.

## 7. Discussion

It is acknowledged that the findings of this research are specific to the case of BMIC project. Nevertheless, generalizable results are possible by the use a historical data of land transactions, where available, that contains information on the change of land use, time of change and value of land before and after. To the best of the author's knowledge, such data is currently unavailable for Indian cities. This research is an important step towards more nuanced methods of estimation of the market value of land.

Table 6 presents estimated value of compensation for compulsorily acquired land under (i) KIAD act of 1966, which refers to the Land Acquisition Act 1894 for compensation determination; (ii) the new land acquisition act of Right to Fair Compensation and Transparency in Land Acquisition, Rehabilitation and Resettlement Act, 2013 and (iii) compensation incorporating blight and hope value:

**Table 6.** Comparing compensation under the current acts and research findings.

| Components of Compensation | Fair Market Value (KIAD Act and Land Acquisition Act of 1894) | Compensation Based on Land Acquisition Act 2015 | Compensation Based on Analysis of BMIC Project |
| --- | --- | --- | --- |
| Market value (MV) of Land in its current use | MV | MV | MV |
| Blight | | | 0.31 MV |
| Hope Value | | | 2.39 MV to 8.35 MV |
| Solatium | 0.3 MV | MV to 3 MV | |
| Total (MV + Blight + Hope value + Solatium) | 1.3 MV | 2 MV to 4 MV | 3.7 MV to 9.66 MV |

Source: Authors.

Table 6 demonstrates that the current and proposed compensation mechanisms inadequately compensate for the loss of land. Fair compensation should account for 'hope' value and blight in calculating the compensation payable to landlords.

This research acknowledges that these results will vary on a case-to-case basis for each project of compulsory acquisition of land in different jurisdictions. The methodological contribution of this research it the innovative application duration model to take account of futuristic changes in land uses and values. This method can be used to develop a nuanced model of estimation of land value that can take account of its futuristic development potentials.

## 8. Conclusions

Compulsory land acquisition is a contentious issue between landowners and acquirers. Despite the underlying principle for the legal framework being 'just' or 'fair' compensation to the affected landowners, in practice, 'fair' compensation is narrowly equated to the current 'market value' of land. The estimation of 'market' value does not account for blight (loss of value caused by acquisition notice) and 'hope' value.

Using the BMIC project in Bengaluru, this paper estimates the hope value and blight. Results obtained for the BMIC project in Bengaluru suggest that land value depreciates by 31 per cent due to the negative impact of acquisition notification. Taking base value of non-notified land under agricultural use, in an agricultural village, the hope value ranges between 2.39 to 8.35 times the market value, depending upon the use of land and village level activities. Also, there is caste-based discrimination in the land market, and land ownership is significant for social equality and the empowerment of the weaker segments. These results indicate that the KIAD Act of 1966, under which land has been acquired, clearly under-compensated the landowners. Even the new Act (LAA 2013), which proposes compensation to the extent of four times that market value of land, under-compensates in some circumstances.

In summary, research findings suggest a significant positive impact of an infrastructure project on the development potential of land in its vicinity, which significantly improves the value of land. The futuristic value appreciation due to the project, and other unforeseeable factors, accrues to those who continue owning land in the project catchment area. On the contrary, those who lose land in the process of compulsory acquisition miss out the benefits.

In addition to losing out on the economic value of futuristic development potential of land, many landowners also experience the loss of value due to reduction consequential to prolonged notification for acquisition of land or 'blight'. These losses build up the financial hardships of affected landowners, particularly those who are in need to sell or mortgage their land during the notification period.

A significant negative impact on property value is observed for land parcels owned by marginalized segments of Scheduled Castes and Scheduled Tribes. Further research is demanded to understand the causes of value reduction associated with the caste of the landowner.

The above findings are a helpful guide in designing a fairer compensation mechanism that encapsulates these financial losses, which are definite yet not compensable unless negotiated in court—hope value and blight. A noteworthy contribution of this research is the methodology designed through the amalgamation of duration model and hedonic price model for determining the hope value and blight. This contributes to the discussion on improving the standard valuation methods to take account of hope value where relevant.

**Author Contributions:** J.S.: Conceptualisation (50%), methodology (50%), Analysis (50%) and Writing (70%). P.T.: Conceptualisation (50%), methodology (50%), analysis (50%), Writing (30%)**.** All authors have read and agreed to the published version of the manuscript.

**Funding:** The study received no external funding.

**Institutional Review Board Statement:** The study uses secondary data and human ethics approval was not required.

**Data Availability Statement:** The data is not publicly available but can be obtained from Sub-Registrar Office, Bengaluru on request for research subject to their approval.

**Conflicts of Interest:** The authors declare no conflict of interest.

## Notes

1.  Functioning is defined as the state of being or doing. In simple words, functionings are the usefulness derived from the resources which one has access to. Ownership of land offers many functionings to its owners, as discussed by Rao [24,25,26].
2.  Refer to Sams [31,32,33], Singh [34] and Rao [2] and Wahi et al. [35] for more discussions on legal disputes on the compulsory acquisition of land in Australia and India.
3.  Sams [31] highlight valuation challenges associated with 'no-scheme' assumption.
4.  Prior to 2006, land transactions were recorded manually by SROs and each transaction has a physical file stored at SRO where property was registered. These records are difficult to access.

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
