# Peer review of "Measuring Inadequacy in Compensation for the Compulsory Acquisition of Land: Evidence from Bengaluru, India"

_land, doi:10.3390/land11050664_

Round 1

Reviewer 1 Report

See attached file

Author Response

Reviewer 1

Review of CP Karnataka
Interesting research, some suggestions offered for improved presentation:

  1. a)  ; also.
  2. b)  References are all Indian, with exception of Scotland and Olanrele. Any other relevant literature on other countries?

Response: We have now added new international literature throughout the article and expanded the literature section. Key mentions are Sams, 2015; Drapikovskyi, et al., 2020; Grzesik & Źróbek, 2017; Rao, et al., 2020

  1. c)  Provide flowchart of procedure, including dispute resolution when compensation is disputed. Any difference in practice in other jurisdictions, particularly other Indian states?

Response: We have introduced a new reference to guide readers to suitable reference that explains the process as well as the issues of unfairness. Section 1 papa 2 is modified to state: “An avid reader may refer to Shukla (2021) for a detailed understanding of the process of compulsory acquisition in India and issues of unfairness.” At a broad level the process of compulsory acquisition is comparable across states, with slight variations. It is beyond the scope of this research to compare the process across states.

  1. d)  Add map (if possible) of location of notified and not-notified cases, also data on size range (land areas and compensation figures) of cases.

Response: Unfortunately, map is not available for the study area. Also, there is limited information available on acquired land parcels. Nevertheless, we acknowledge reviewer’s comment and have added the following information under Section 3 on study area:

“A total of 23,846 acres of land is proposed to be acquired for the project from across eight different Talukas (or administrative sub-divisions within a city) of Bangalore North (885 acres), Bangalore South (5,089 acres), Ramnagaram (8,170 acres), Channapatna (3,572 acres), Maddur *481 acres), Mandya (667 acres), Srirangapatnam (4,839 acres), and Mysore (173 acres) (KPWD, 1998)”

Recommend further discussion and comment on following:

  1. a)  Length of time between notification and final settlement, which appear excessive. How many cases are on-going? Are authors directly involved as experts in individual disputes? Is reasoning made explicit in determinations?

Response: We have modified section 3 to incorporate reviewer’s valuable comments:

“Through a series of acquisition notifications between 1996 and 2004, Karnataka Industrial Areas Development Board (KIADB) had notified all land for compulsory acquisition under the Karnataka Industrial Areas Development (KIAD) Act of 1966.” Information on number of ongoing cases is not available. However, concluded court case reports from the High Court and Supreme Court of India is available in the public domain. Authors are not involved in individual disputes but could extract useful information about the project from the case reports.

  1. b)  Reasons for huge variation in hope values (4%-409%).

Response: This range was incorrectly mentioned in the paper and been updated to 2.39 to 8.35 times the market value of agricultural land in 2006. The reason for this variation is that hope value depends on the combination of development potential of the village and the land parcel. Therefore, depending on the location (village) and type of improved land use, the hope value will vary.  To explain this, we have now mentioned under Section 7 (Discussion) – “Taking base value of non-notified land under agricultural use, in an agricultural village, the hope value ranges between 2.39 to 8.35 times the market value, depending upon the use of land and village level activities.”

  1. c)  Political involvement and different actors in compensation disputes.

Response: Reviewers are rightly understanding that there is political involvement and multiple stakeholders. There is extensive discussion in court case reports on the ulterior motives of the ruling party and private partners involved in the project. We acknowledge that these discussions are beyond the scope of this paper.  

  1. d)  Any specific proposals to change law/practice in valuation methods and procedures?

Response: Conclusion is modified to state “The above findings are a helpful guide in designing a fairer compensation mechanism that encapsulates these financial losses, which are definite yet not compensable unless negotiated in court – hope value and blight. A noteworthy contribution of this research is the methodology designed through the amalgamation of duration model and hedonic price model for determining the hope value and blight. This contributes to the discussion on improving the standard valuation methods to take account of hope value where relevant.”

Reviewer 2 Report

The article addresses important issues of insufficient compensation in the event of land acquisition (expropriation) for investment purposes. In particular, the issues of considering potential future changes in land value resulting from possible changes in land use are described. The topic undertaken by the authors is a frequent topic of discussion in many countries. How should the purchase price be shaped in such cases? What factors should be considered? Is it just the market value or also other factors?

The introduction by far lacks an overview of the approaches used in many different countries in similar cases. How is land for large investments, especially public ones, purchased? How is the land value determined? Are these negotiations with owners or just a market valuation? Are appropriate additional compensations added to market prices, in what amount and on what do they depend?

The article has a separate chapter "Literature", but it is based only on research (Olanrele et al., 2017). Some information is provided for the United Kingdom, Denmark, the United States and New Zealand. This is definitely too little and requires supplementation, especially since the subject is present in the existing research. The concept of "The Values ​​of Hope" was based on Rao's research and experiences from India and Scotland.

Please describe briefly the regulations for estimating the market value of land in India.

Chapter 3  (“About the Bangalore-Mysore Infrastructure Corridor (BMIC) project”) the correct name is "Study Area"?

The entirety of Chapter 4 should be included in the Methodology chapter.

Similarly, in chapter 6 (6. Estimation Results), the name "Results" is sufficient.

Chapter 6.2. “Hedonic function for land values: and table 2. The proposed function formulas have not been formulated. This chapter contains a table of function coefficients, not the function itself, so the description of the table is not correct.

In table 6 there is the designation MV, no explanation. This is probably short for Market Value, but that shouldn't be a matter of the reader 's guesswork.

Why are Tables 5 and 6 located in the discussion? There are results there, important for the discussion, but they are calculation results nonetheless. They should be placed in the "Results" chapter

Is India's 30% more than market price some form of taking into account the assumption of increasing the value of land above market value?

What are the authors' suggestions for solutions? The very high value of "Hope value" in the studied area, compared to the market price, has been proven. But there was historical data showing rising prices and changes in land use. How to calculate 'Hope value' at redemption when this type of data is not known? Each area type can have completely different coefficient values. The values ​​obtained in the study cannot be used for other cases.

Author Response

Reviewer 2

The article addresses important issues of insufficient compensation in the event of land acquisition (expropriation) for investment purposes. In particular, the issues of considering potential future changes in land value resulting from possible changes in land use are described. The topic undertaken by the authors is a frequent topic of discussion in many countries. How should the purchase price be shaped in such cases? What factors should be considered? Is it just the market value or also other factors?

The introduction by far lacks an overview of the approaches used in many different countries in similar cases. How is land for large investments, especially public ones, purchased? How is the land value determined? Are these negotiations with owners or just a market valuation?

Response: In response the reviewer’s comments, authors’ have made modification under different section of the paper. For example:

  1. Author’s included following statements to define the scope of this paper: “This research does not question the use of powers of compulsory acquisition and instead argues for a fair compensation that can take account of ‘hope value’ and ‘blight’ currently remain uncompensated in many countries including India. Negative impact of ‘blight’ and the problem of measuring the impact is specific to the case of compulsory acquisition of land. However, the problem of estimating the ‘hope value’ is widely acknowledged in the valuation profession (Drapikovskyi, et al., 2020). For example, global valuation standards by IVCS (2022) and RICS (2022) consider it necessary to take account of the likelihood of land use/zoning to change in the future. Thus, while feeding into the bigger objective of designing a fairer compensation mechanism, this research makes methodological contribution into estimation of ‘hope value’. “
  2. A footnote on the definition of market value in India in included.
  3. Article acknowledges: “There is ample empirical evidence from many parts of the world, including India, to suggest that the majority of legal disputes for compensation conclude in favour of the landowner and upward improvement in compensation is commonly observed (Wahi et al., 2017; Newell et al., 2011; Singh, 2012).”
  4. To explain the process of compulsory acquisition and dispute resolution, we have modified the introduction to include reference for further reading: “An avid reader may refer to Shukla (2021) for a detailed understanding of the process of compulsory acquisition in India and issues of unfairness.”

Are appropriate additional compensations added to market prices, in what amount and on what do they depend?

Response: Section 2 is modified to state “…components of compensation in the UK, Denmark, US and New Zealand, though described differently, broadly include the market value of land and improvements; severance; disturbance; and injurious affection (refer Olanrele, et al., 2017 for details on components of compensation across different countries). In addition to the above losses, the Indian legislature (Land Acquisition Act, 1894) adds to the market value of land in current use another 30% solatium as compensation in lieu of compulsory nature of sale (refer section 23(2), Land Acquisition Act of 1894). A similar justification is used by other countries like Australia and Scotland to offer additional solatium on top of market value of property particularly when the primary residence of a person is to be acquired.”

The article has a separate chapter "Literature", but it is based only on research (Olanrele et al., 2017). Some information is provided for the United Kingdom, Denmark, the United States and New Zealand. This is definitely too little and requires supplementation, especially since the subject is present in the existing research. The concept of "The Values ​​of Hope" was based on Rao's research and experiences from India and Scotland.

Response:

We have now added new international literature on throughout the article and in the Literature. These include: Sams, 2015; Drapikovskyi, et al., 2020; Grzesik & Źróbek, 2017; Rao, et al., 2020

A new paragraph in also added to the Literature: “As mentioned by Drapikovskyi, et al. (2020), option pricing models such as binomial model, Black-Scholes model, and the Samuelson-McKean model, are most popularly employed in determining the hope value of land. However, option pricing models are not dynamic and do not take account of varying probability of change of land uses over time. To overcome this problem, this reseach innovatively uses the duration model together with the traditional hedonic price model, as discussed under Section 4. Findings from this research measure these losses in the Indian context and provide guidance on methodology using an innovative amalgamation of duration model and hedonic price model in determining the above (apparent) losses that deserve compensation.”

Please describe briefly the regulations for estimating the market value of land in India.

Response: This is a very useful comment. We have modified literature to say: “Under Section 26 of the Right to Fair Compensation and Transparency in Land Acquisition, Rehabilitation and Resettlement Act, 2013, the market value of land in India is defined as the higher among the following: (a) the transaction value specified in the Indian Stamp Act, 1899 (2 of 1899); or (b) the average sale price for similar type of land situated in the nearest village or nearest vicinity area, during immediately preceding three years of the year in which such acquisition of land is proposed to be made.”

Chapter 3  (“About the Bangalore-Mysore Infrastructure Corridor (BMIC) project”) the correct name is "Study Area"?

Response: Agreed. Sub-title is changed to “Study area: Bangalore-Mysore Infrastructure Corridor (BMIC) project”

The entirety of Chapter 4 should be included in the Methodology chapter.

 Response: Agreed. Section 4 is indeed the methodology. To explain the content of methodology section to the reader, authors we are using the title “A theoretical framework for estimation: The ‘hedonic’ approach”

Similarly, in chapter 6 (6. Estimation Results), the name "Results" is sufficient.

 Response: Agreed. Sub-title is changed to “Results”

Chapter 6.2. “Hedonic function for land values: and table 2. The proposed function formulas have not been formulated. This chapter contains a table of function coefficients, not the function itself, so the description of the table is not correct.

 Response: Agreed. Table caption is changed to:  Alternative hedonic price model estimation results for land value.

In table 6 there is the designation MV, no explanation. This is probably short for Market Value, but that shouldn't be a matter of the reader 's guesswork.

Response: Agreed. Appropriate change is made to the table.

Why are Tables 5 and 6 located in the discussion? There are results there, important for the discussion, but they are calculation results nonetheless. They should be placed in the "Results" chapter

Response: Discussion is on results and table 5 and 6 are better positioned in this section as these tables support the discussions.

Is India's 30% more than market price some form of taking into account the assumption of increasing the value of land above market value?

Response: The additional payment of 30% is mentioned as solatium and is interpreted as the compensation for the forceful nature of sale. This is explained in following words in the paper: “In addition to the above losses, the Indian legislature (Land Acquisition Act, 1894) adds to the market value of land in current use another 30% solatium as compensation in lieu of compulsory nature of sale (refer section 23(2), Land Acquisition Act of 1894).A similar justification is used by other countries like Australia and Scotland to offer additional solatium on top of market value of property particularly when the primary residence of a person is to be acquired.

While it can be argued that this additional payment accounts of the assumption of increasing land values over time, it raises questions on the comprehensiveness of market value and the methods of valuation - Why do market value estimations not take account of this future value appreciation even though the definition states “market value is the present value of sum of all rents in the future”?

What are the authors' suggestions for solutions? The very high value of "Hope value" in the studied area, compared to the market price, has been proven. But there was historical data showing rising prices and changes in land use. How to calculate 'Hope value' at redemption when this type of data is not known? Each area type can have completely different coefficient values. The values ​​obtained in the study cannot be used for other cases.

Author’s response:

There, is further work required on the topic. The contribution of this research is twofold: firstly, to innovatively use hazard function to estimate the probability of change of land-use and its impact on value; and secondly to contribute to the ongoing debate on inadequate measure of market value of land and consequential inadequate compensation in cases of compulsory acquisition of land.

We acknowledge that these results will vary on a case-to-case basis for each project of compulsory acquisition of land. The methodological contribution of this research it the innovative application hazard risk model to take account of futuristic changes in land uses and values. This method can be used to develop a nuanced model of estimation of land value that can take account of its futuristic development potentials. This research is an initial step towards this greater objective.

It is acknowledged that the findings of this research are specific to the case of BMIC project. Nevertheless, one can derive generalisable findings by using a historical data of land transactions, where available, that contains information on the change of land use, time of change, and value of land before and after. To author’s best knowledge, such a data is currently unavailable for Indian cities. This research is an initial step towards more nuanced methods of estimation of the market value of land.

Reviewer 3 Report

Dear authors,

nice job!

I attach my comments below.

Author Response

Reviewer 3

Dear authors, nice job!

First, set the Land Journal to the correct format.

.

You mentioned the BMIC project, but you talk about "as observed in the BMIC Project" you would have to explain that more.

Response: Agreed. To avoid confusion, we have removed "as observed in the BMIC Project".

You have referred to 'blight' twice, but it is explained here. International readers may be confused.

Response: Agreed. To avoid confusion, we have now stated “explained under Section 2” when using it the first time under Section 1.

The above text is very good, try to avoid the apostrophe.

Response: It was difficult to understand the above comment without reference to the line number. Nevertheless, we have undertaken a through proofreading and assume the above problems are resolved throughout the manuscript.

  1. About the Bangalore-Mysore Infrastructure Corridor (BMIC) project

I recommend moving this point to the introduction. Right here you explain some things you mentioned above.

Response: We havemodified the sub-title of this section to “Study area: Bangalore-Mysore Infrastructure Corridor (BMIC) project”. This information is now appropriately positioned in the paper.

It is necessary to insert a map with the state of Karnataka (India), Bangalore and Mysore.

Response: Unfortunately, maps are not available for the study area.

For this research, we refer to the economic we estimate alternative forms of Box-Cox

We estimate the following functional forms

Please, change to impersonal writing form

Response: Agreed and resolved at most locations.

  1. Estimation Results

Blight due to acquisition notification: Sometimes, there is a time gap between the

actual acquisition of land and its notification for acquisition, as observed in the

BMIC Project

Blight is defined as the depreciation in property value consequential to a notification

for compulsory acquisition for a public project. For example, when a property is

earmarked for a futuristic public purpose

“Value of apartment is not equivalent to the value of land. Hence a hedonic price function for apartment sale value as a function of apartment characteristics and estimated to derive the land value for land under apartment use. We assume that the relevant value for consideration in our apartment. The undivided share of land is defined as the proportionate share of land of an apartment in the land area of the entire apartment complex project. This is calculated by dividing built-up area or saleable floor area of a single apartment by total built up area or saleable floor area of the entire project multiplied by total land area of the project. Estimated function is then used to predict value of undivided share of land for all observations pertaining to apartment sales in the data. This allows us to estimate hedonic price function for land for all apartment sales. In the second step hedonic functions for land sales have been estimated. “

Please explain this paragraph better, because suddenly "apartment" becomes relevant

Check the apostrophes.
You can insert examples like complex land (apartment + agricultural activities)... Table 1. why these variables, support more

Response: Agreed. We have modified Section 6 to say: “Land transaction data was inclusive of transaction of apartments. Information on value of land used for apartment, referred to as the undivided share of land (USL), is crucial to this research. Therefore, we used a hedonic price model to extract the value of USL. Hedonic function considered the transaction value of apartment as a function of USL and other attributes presented in Table 1.”

You mentioned that you have a location. You could include in the discussion of spatial correlation. Contour lines of land value illustrating the effects of location and georeferenced explanatory variables as next steps in your research. This might increase readability.

Response: This is a good suggestion. However, the study area is a plain land with slight variation in contours. Therefore, contour may not have a significant impact on land value.

You can review
Kennedy et al. 2002; King y Schreiner 2004; Snyder et al. 2007, 2008; Zhou 2010

Reviewer 4 Report

The Abstract is clear and captures the main ideas and key findings of the paper.
The Introduction leads the reader to the main purpose of the paper. The Literature section should have reviewed the state of the art on the topic at stake. However, the literature review is missing.
The Results focus too much on presenting the model results, too extensively (tables 2 and 3). I think that relating the results with the driving factors is sufficient.
It would be useful for the paper to have few discussions on how the (expected) results relates to the existing literature.

Author Response

Reviewer 4

The Abstract is clear and captures the main ideas and key findings of the paper.
The Introduction leads the reader to the main purpose of the paper.

The Literature section should have reviewed the state of the art on the topic at stake. However, the literature review is missing.

Response:

Response: We have now added new international literature throughout the article and expanded the literature section.

The Results focus too much on presenting the model results, too extensively (tables 2 and 3). I think that relating the results with the driving factors is sufficient.

Response: An important objective of this research is to make methodological contribution to estimating hope value. Given the complexity of using an innovative methodology by combining the duration model and hedonic price model, we feel the need to explain the results adequately.

It would be useful for the paper to have few discussions on how the (expected) results relates to the existing literature.

Response: We have added the following to acknowledge that results are not generalisable and therefore not comparable to existing literature:

“This research acknowledges that these results will vary on a case-to-case basis for each project of compulsory acquisition of land in different jurisdictions. The methodological contribution of this research it the innovative application duration model to take account of futuristic changes in land uses and values. This method can be used to develop a nuanced model of estimation of land value that can take account of its futuristic development potentials. This research is an initial step towards this greater objective.”

Round 2

Reviewer 2 Report

I looked carefully at all changes made by the authors, as well as all responses to comments made in the reviews.
In my opinion, the article is of high scientific quality in its current form and can be published in the Land journal.
I only have reservations about the list of references, some are numbered, some are not.

Author Response

The references have been verified for their relevance to the paper.